# Therapeutic Strategies in Children with Epilepsy: A Quality-of-Life-Related Perspective

**DOI:** 10.3390/jcm13020405

**Published:** 2024-01-11

**Authors:** Hideaki Kanemura

**Affiliations:** Department of Pediatrics, Toho University Medical Center Sakura Hospital, 564-1 Shimoshizu, Sakura 285-8741, Chiba, Japan; hideaki.kanemura@med.toho-u.ac.jp; Tel.: +81-43-462-8811

**Keywords:** frontal, prefrontal, behavior, stigma, seizure severity, interictal epileptiform discharge (IED), quality of life (QOL)

## Abstract

**Back ground:** Children with epilepsy are affected by several factors, including clinical and social variables. Among these variables, cognitive decline and behavioral disturbances, perceptions of stigma, and fatigue can lead to reductions in quality of life (QOL). Epileptic activities, including seizure severity, frequent seizures, and status epilepticus (SE), have been identified as important predictors of QOL. In addition, the frequency of interictal epileptiform discharges (IEDs) on electroencephalogram (EEG) may also be an important predictor of QOL, because IEDs can lead to cognitive decline and behavioral disturbances. Moreover, frequent seizures and/or IEDs may play a role in emotional mediators, such as stigma and fatigue, in childhood epilepsy. Seizure severity and/or IEDs are, therefore, important QOL-related factors in childhood epilepsy. **Seizure severity as a QOL-related factor:** Frontal lobe dysfunctions, such as cognitive decline and behavioral disturbances, can result in reduced QOL for both the child and their family. Frontal and prefrontal lobe growth disturbances can be present during active-phase epilepsy in some children with neuropsychological impairments. Recovery from prefrontal lobe growth disturbances may depend on the active seizure period. Children with a shorter active seizure period can recover from disturbances in prefrontal lobe growth more rapidly. In contrast, recovery may be delayed in children with a longer active seizure period. Moreover, frequent seizures can lead to seizure-associated headaches, perceptions of self-stigma and parental stigma, and fatigue. Accordingly, severe seizures can lead to neuropsychological impairments in association with prefrontal lobe growth disturbances in children with epilepsy. **EEG abnormalities as QOL-related factors:** IEDs on EEG, representing persistent pathological neuronal discharges, may be associated with several pathological aspects. Frontal IEDs can be a risk factor for recurrent seizures, cognitive decline, and behavioral disturbances, and they may also play a role as emotional mediators similar to stigma. In addition, behavioral disturbances may result in the presence of secondary bilateral synchrony (SBS) on EEG. Behavioral disturbances can be improved in association with a reduction in IEDs in children with frontal IEDs and SBS. Therefore, EEG abnormalities, such as frontal IEDs and SBS, can also lead to neuropsychological impairments in children with epilepsy. **Therapeutic strategies in children with epilepsy:** Seizure severity and IEDs on EEG may be associated with neuropsychological impairments, leading to QOL reduction. Therapeutic management may be desirable to reduce seizures and EEG abnormalities, such as frontal IEDs and SBS, as early as possible to improve QOL in children with epilepsy. During antiseizure medication (ASM) selection and adjustment, physicians should strategize the therapeutic approach to controlling seizures and suppressing EEG abnormalities in children with epilepsy. Among various ASMs, novel ASMs, such as levetiracetam and perampanel, may suppress both clinical seizures and IEDs on EEG; thus, these novel ASMs may represent an important addition to the treatments available for epileptic children presenting with frontal IEDs and SBS.

## 1. Introduction

Quality of life (QOL) in children with epilepsy seems to be influenced by various factors, including clinical and social variables. Among these variables, cognitive decline and behavioral disturbances, perceptions of stigma, and fatigue can lead to reductions in QOL. These disturbances may be associated with epileptic activities. Thus, seizure control is critical because of the improvement in QOL [1].

Among various epileptic activities, seizure severity, frequent seizures, and the presence of status epilepticus (SE) are regarded as the main QOL-related factors. Seizure clusters may be associated with higher seizure frequency, higher risk of treatment resistance, and lower likelihood of seizure remission, according to a systematic review [2]. In children with seizure clusters, QOL could be lower than that in children with isolated seizures [2]. Seizure clusters may also adversely affect the productivity of patients and their caregivers [2]. In addition, frequent seizures and SE can lead to neuropsychological impairments [3,4]. Accordingly, seizure severity may be associated with a reduction in QOL in children with epilepsy. On the other hand, electroencephalogram (EEG) findings, such as the frequency of interictal epileptiform discharges (IEDs), which are among the epileptic activities, may also be related to cognitive decline and behavioral disturbances [5]. Like frequent seizures, IEDs on EEG could also mediate emotional states, including stigma and fatigue [6]. These findings suggest that epileptic activities, including frequent seizures and the presence of SE and IEDs on EEG, may lead to reduced QOL in children with epilepsy.

On the other hand, stigma is regarded as an important QOL-related factor in epilepsy. The perception of stigma related to the epileptic condition may be very severe and is often under-recognized by clinicians [7]. Stigma has a marked negative impact, not only for children but also their families. Reducing perceptions of stigma is, therefore, necessary for the clinical management of childhood epilepsy.

In addition, fatigue is also associated with reduced QOL in patients with epilepsy. Fatigue may correlate with psychosocial factors, including anxiety, depression, and sleep problems [8]. In addition to these variables, epileptic activities may be associated with fatigue. A previous study showed that frequent seizures can lead to increased fatigue in epileptic patients [9]. Thus, fatigue may also relate to epileptic activities.

Neuropsychological impairments in children with epilepsy, such as cognitive decline, behavioral disturbances, perception of stigma, and fatigue, that relate to frontal lobe dysfunction can be caused by several factors. Frontal lobe lesions may lead to these disturbances in consonance with lesion location. However, children with no lesions can also present with these disturbances. A previous study showed a negative correlation between the frequency of IEDs on EEG and cognitive function in self-limited epilepsy with centrotemporal spike (SeLECTS) [10]. These disturbances may thus relate to epileptic activities and topography. The frontal lobes mature over a long period and so are easily damaged by various factors. Damage to the frontal regions during childhood interferes with maturational and organizational processes, which can lead to neuropsychological impairments [11]. Results from previous investigations have suggested that severe seizures, as reflected by certain statuses, such as frequent seizures and SE, can impair the developing brain [11,12,13]. In combination with these studies, seizure severity and IEDs on EEG may lead to cognitive, behavioral, and psychological disturbances. Epileptic activities, such as seizure severity and/or IEDs, are important QOL-related factors in childhood epilepsy.

We previously investigated the association between QOL-related factors and epileptic activities. The objective of this review was to assess the association between epileptic activities and QOL-related factors, focusing on cognitive and behavioral disturbances, stigma, and fatigue, according to our research findings. Based on these observations, therapeutic strategies in children with epilepsy are discussed in the following sections.

## 2. Can Seizure Severity Lead to Reduced QOL in Childhood Epilepsy?

### 2.1. Seizure Severity and Cognitive/Behavioral Disturbances: Are They Related?

Neuropsychological impairments, such as cognitive decline and behavioral disturbances, reduce QOL in childhood epilepsy and may be induced by frontal lobe dysfunction. In addition, QOL reduction in children can also reduce QOL in the family. Thus, frontal lobe dysfunction can result in reduced QOL for both the child and their family.

The frontal lobes are the largest cortical regions of the brain, comprising approximately 40% of the cerebral cortex. Among these regions, the prefrontal regions involve wide networks [14]. Because of these connections, the prefrontal cortex can receive abundant information from all parts of the cerebrum and can affect information processing in those parts. Prefrontal lobe neurons and glial cells are readily influenced by various factors, so prefrontal lobe functions are regarded as being vulnerable for a long period [15]. Accordingly, severe seizures, such as frequent or prolonged seizures, result in negative effects on prefrontal lobe functions more easily than on other cortical regions [14,15]. Considering these findings, epilepsy associated with prefrontal regions in children may be associated with several neuropsychological impairments in comparison with healthy subjects [3].

In other cerebral regions, seizures can result in memory and learning disturbances, which relate to temporal lobe dysfunctions [16]. Temporal lobe seizures can also lead to behavioral disturbances [17]. However, direct relationships between seizures and temporal or other cerebral lobe functioning have not been fully revealed. Further research is needed to discuss these aspects.

#### 2.1.1. Prefrontal Lobe Growth in Frontal Lobe Epilepsy

Understanding how frontal lobe epilepsy (FLE) impacts the life of patients is important. In a serial three-dimensional (3D) magnetic resonance imaging (MRI) volumetric study, the growth of the frontal and prefrontal lobes in children with drug-responsive FLE without neuropsychological impairments was similar to that in healthy subjects [3]. In contrast, frontal and prefrontal lobe growth disturbances were present during the active epileptic phase in refractory FLE patients with cognitive decline and behavioral disturbances. However, a difference associated with the active seizure period was present. A short active seizure period was associated with prompt growth recovery. In children with a longer active seizure period, the growth disturbance was more severe, and growth recovery took longer [3] (Table 1). Frequent seizures in children with FLE may thus induce prefrontal lobe growth disturbances, which can lead to neuropsychological impairments.

#### 2.1.2. Prefrontal Lobe Growth in SeLECTS

SeLECTS is considered a condition free of neurological and psychological impairments. However, children with SeLECTS sometimes present with severe aggravation of epileptic manifestations, cognitive decline, and behavioral disturbances [18]. Frontal and prefrontal lobe volumes and, in particular, the prefrontal-to-frontal lobe volume ratio showed growth disturbances during the active seizure period in patients presenting with atypical evolution [14]. However, differences associated with the active seizure period were present. The active seizure period was shorter in patients with prompt growth recovery. In patients with a longer active seizure period, growth disturbances were more severe, and, again, growth recovery took longer in these patients [14] (Table 1). Seizure severity in SeLECTS may also be associated with prefrontal lobe growth disturbances, again leading to neuropsychological impairments.

#### 2.1.3. Prefrontal Lobe Growth in Self-Limited Epilepsy with Autonomic Seizures

Self-limited epilepsy with autonomic seizures (SeLEAS), which represents Panayiotopoulos syndrome, is generally accepted as lacking neuropsychological impairments. However, cognitive decline and behavioral disturbances may be present in at least some children with SeLEAS. SE can induce cerebral damage to various degrees. In SeLEAS patients, seizures tend to be prolonged, with subsequent focal or focal-to-bilateral tonic–clonic SE [4]. A sequential study using 3D–MRI volumetry showed that frontal and prefrontal lobe growth disturbances were present after episodes of SE in SeLEAS patients presenting with behavioral disturbances. In a patient with only one episode of SE, growth disturbances soon recovered. Conversely, the recovery of growth ratios was delayed in patients with several episodes of SE [4] (Table 1). Moreover, the cognitive scores, as measured using the Wechsler intelligence scale for children, dropped after the SE episodes [4]. The presence of SE in children with SeLEAS may thus induce growth disturbances in the prefrontal lobe, which can lead to neuropsychological impairments.

### 2.2. QOL-Related Factors: Headache

Epilepsy and migraine are part of a heterogeneous family of neurological disorders [19]. The prevalence of headache is extremely high, so concomitant headache can be present in many patients with epilepsy. Approximately 35% of epileptic children experienced headaches in association with seizures in our previous study [20]. The frequencies of seizures in children with and without seizure-associated headache were 4.1 and 1.3 times per year, respectively [20] (Table 2). Thus, seizure recurrence can induce headaches in association with seizures, which leads to reduced QOL in children with epilepsy.

### 2.3. QOL-Related Factors: Fatigue

Fatigue has a negative impact on QOL in patients with various chronic diseases, including epilepsy [21,22,23]. Our previous study showed that the mean Fatigue Severity Scale scores in epileptic children were significantly higher than those in non-epileptic children [24]. The frequency of seizures was identified as the only significant clinical manifestation associated with fatigue using multiple linear regression analysis. Moreover, children with frequent seizures presented with more severe fatigue [24] (Table 2). Accordingly, frequent seizures can lead to the presence of fatigue in children with epilepsy.

### 2.4. QOL-Related Factors: Perception of Stigma among Children

The perception of stigma among epileptic patients is a negative psychological issue associated with a reduction in QOL. Stigma may be considered one of the psychological factors that affect QOL, along with seizure factors. Stigma has a negative effect on self-esteem and social status, thus leading to a poor prognosis, including isolation and delayed initiation of treatment for epileptic patients [25]. Frequent seizures could lead to psychosocial impairments in children. Our previous study, which used the Child Stigma Scale, showed that the perceptions of stigma in association with seizure frequency were severe (Table 2) [26]. Thus, stigma has negative effects on social identity in children with epilepsy who experience frequent seizures.

### 2.5. QOL-Related Factors: Perception of Stigma among Parents

Epilepsy in children can be a risk factor for stress in their parents [27,28,29,30]. Parents of children with intractable epilepsy tend to experience severe anxiety in relation to recurrent seizures, and this parental state of anxiety can lead to poor adaptive function in children [31]. Frequent seizures are, therefore, an important issue with respect to parenting stress [32]. The parents of children with epilepsy showed higher scores on the Parent Stigma Scale than the parents of healthy children [33]. Moreover, greater perceptions of stigma among parents correlated with higher seizure frequency [33] (Table 2). Accordingly, frequent seizures in children with epilepsy can lead to greater perceptions of stigma among parents.

However, the relationship between seizures and stigma has only been analyzed individually without simultaneously considering other biopsychosocial factors; therefore, there may be limitations in the understanding of this relationship.

## 3. Can EEG Abnormalities Lead to QOL Reduction in Children with Epilepsy?

### 3.1. Association between IEDs on EEG and Seizure Recurrence

EEG abnormalities, such as IEDs, can be conceptualized as pathological neuronal discharges [34]. This can lead to the fact that IEDs on EEG are associated with seizure recurrence. In SeLECTS, recurrent seizures and prolonged periods of frequent IEDs were correlated [35]. This finding suggests that the occurrence of frequent IEDs and the prolongation of this state may lead to recurrent seizures in SeLECTS. In addition, seizure recurrence may be associated with the location of EEG foci. Our study showed that frontal IEDs induced recurrent seizures after a first unprovoked seizure more often than other EEG foci [36] (Table 3). Thus, the frequency of IEDs and frontal IEDs may be associated with recurrent seizures.

### 3.2. Association between IEDs on EEG and Cognitive/Behavioral Disturbances

Cognitive decline and behavioral disturbances are seen more often in children with more severe EEG abnormalities [37,38,39]. Children can exhibit behavioral disturbances in association with IEDs on EEG without clinical seizures [5,40]. These disturbances may be associated with IEDs on EEG.

#### 3.2.1. Association between IEDs and Cognitive/Behavioral Disturbances: SeLECTS

These disturbances were present in relation to a prolonged period of frequent IEDs in SeLECTS [41]. Moreover, these disturbances were also associated with a prolongation of the frontal EEG focus [41] (Table 3). Cognitive function can deteriorate in children with frequent IEDs. In addition, neuropsychological functioning can recover with the normalization of EEG findings [42]. Thus, neuropsychological impairments may be associated with IEDs on EEG in SeLECTS (Table 3). Tassinari et al. reported that the neurophysiological impairments associated with the negative myoclonus or inhibitory seizures seen in atypical SeLECTS suggest the involvement of the frontal cortex, either primarily or secondarily [43]. Moreover, frontal and prefrontal lobe growth disturbances persisted even after seizure disappearance in atypical SeLECTS [14]. These findings suggest that clinical features in atypical SeLECTS may be associated with frontal lobe dysfunctions, which can lead to neuropsychological impairments. In combination with these findings, frequent and frontal IEDs can lead to cognitive decline and behavioral disturbances.

#### 3.2.2. Association between Frontal IEDs and Behavioral Disturbances: ADHD

Frontal IEDs are considered to show various pathogeneses. In children with attention deficit hyperactivity disorder (ADHD) presenting with IEDs, the frequency of IEDs was significantly correlated with the ADHD rating scale (ADHD—RS) score in the frontal IED group, but not in the Rolandic discharge (RD) group [44] (Table 3). In addition, the same study also showed that reductions in IED frequency were significantly correlated with ADHD—RS score reductions in the frontal IED group treated with antiseizure medications (ASMs), but not in the RD group [44] (Table 3). These findings suggest that frontal IEDs can exacerbate behavioral disturbances in children with ADHD.

#### 3.2.3. Association between Frontal IEDs and Behavioral Disturbances: ASD

IEDs in children with autism spectrum disorder (ASD) are frequently located in the frontal region [45]. ASD children with epilepsy with frontal IEDs presented behavioral improvement (reductions in their score according to the Japanese manuals for the Aberrant Behavior Checklist (ABC-J)) in association with EEG improvement (reductions in IED frequency) after ASM treatment [46] (Table 3). This finding suggests that behavioral disturbances can be associated with the frequency of IEDs on EEG and that ASM treatment can lead to both reduced IED frequency and reduced behavioral problems in ASD children with epilepsy with frontal IEDs [46]. Thus, frontal IEDs can lead to an exacerbation of behavioral disturbances in ASD children.

#### 3.2.4. Association between SBS and Behavioral Disturbances: EECSWS

Secondary bilateral synchrony (SBS) on EEG may be associated with cognitive decline and behavioral disturbances. The majority of children with epileptic encephalopathy with continuous spikes and waves during slow sleep (EECSWS), as a representative epileptic syndrome of SBS, present neuropsychological impairments. Our previous study showed that the volumes of frontal and prefrontal lobes were smaller in EECSWS children compared with those in healthy children [47] (Table 3). Moreover, prefrontal lobe growth disturbances were prolonged in children with longer CSWS periods in comparison with those with shorter CSWS periods [47] (Table 3). These findings suggest that persistent, severe abnormalities on EEG may induce prefrontal lobe growth disturbances, which can lead to neuropsychological impairments.

From a therapeutic perspective, the association between the reduction in IED frequency and behavioral improvement after ASM treatment was evaluated in epileptic patients presenting with SBS [48]. Scores for behavioral disturbances measured using the ABC-J were decreased in both EEG responders and EEG non-responders following ASM treatment. Moreover, reductions in ABC-J scores were significantly better in EEG responders than in non-responders. These findings suggest that EEG findings, such as SBS, can lead to behavioral disturbances and that behavioral improvements may be achievable in association with EEG improvement [48] (Table 3). Thus, SBS can lead to behavioral disturbances.

### 3.3. Association between EEG Abnormalities and Stigma

The emotional state can be influenced by various epileptic activities, including EEG abnormalities. A previous study showed that frontal EEG abnormalities might affect neuropsychological functions [49]. Our previous study indicated an association between EEG abnormalities and perceived stigma [50]. Children with frontal IEDs had higher scores on the Child Stigma Scale than those with IEDs in other regions. This finding suggests that frontal IEDs may be associated with a greater perception of stigma. Thus, frontal IEDs may play the role of emotional mediator in certain factors, such as stigma [50] (Table 3). Prefrontal lobe functions are regarded as being vulnerable for a long period. Thus, frontal IEDs may affect frontal lobe functions, such as neuropsychological states, including stigma. However, the mechanisms underlying the association between frontal IEDs and neuropsychological function are unclear. In addition, the relationship between IEDs on EEG and stigma has only been analyzed individually, without simultaneously considering other biopsychosocial factors. Therefore, the relationships among these factors remain unclear.

## 4. How Do We Manage the Treatment of Epilepsy in Children?

### 4.1. Is the Urgent Suppression of Clinical Seizures Needed?

As mentioned above, the presence of frequent seizures and SE can induce growth disturbances in the prefrontal lobe, leading to neuropsychological impairments [3,4] (Figure 1). In addition, recovery from prefrontal lobe growth disturbances may depend on the active seizure period. In several epileptic syndromes, children with a shorter active seizure period can recover from disturbances in prefrontal lobe growth more rapidly. However, such recovery may be delayed in children with a longer active seizure period [3]. These findings support the hypothesized relationship between seizure activities and behavioral disturbances, i.e., “seizure activity per se disrupts behavior”, as suggested by Austin et al. [51]. In addition, SE in children can lead to prefrontal lobe growth disruption. In our volumetric study, more frequent SE episodes led to poorer outcomes [4]. Accordingly, SE can lead to neuropsychological impairments in association with prefrontal lobe growth disruptions (Figure 1). Another study indicated that damage to the frontal regions during childhood can cause deteriorations in neurobehavioral development [52].

Based on these findings, the therapeutic strategy for childhood epilepsy may require seizure remission as soon as possible to prevent neuropsychological impairments.

### 4.2. Is the Urgent Suppression of IEDs on EEG Needed?

As inferred from various studies, frequent IEDs and frontal IEDs can lead to neuropsychological impairments [41] (Figure 1). Reductions in IEDs on EEG may be related to behavioral improvements in ADHD/ASD children with frontal IEDs with or without clinical seizures [44,46,53]. Accordingly, frontal IEDs can lead to neurodevelopmental deterioration in ADHD/ASD, and ASM treatment may be effective for both IED reduction and behavioral improvement in children with ADHD/ASD with frontal IEDs.

With respect to EECSWS-related neurodevelopmental deterioration, previous studies have underlined the parallel course of EECSWS and neuropsychological impairments [54]. Neuropsychological impairments may appear concurrently with EEG abnormalities [55] (Figure 1). Moreover, these impairments may improve concurrently with the disappearance of EEG abnormalities rather than clinical seizures. In children with SBS, behavioral improvements can be associated with EEG improvement [48]. These findings suggest that the active phase of “epilepsy”, not only “clinical seizures”, can be a prognostic factor, and the urgent suppression of IEDs, such as SBS, may thus be warranted to prevent neurodevelopmental deterioration in children presenting with SBS [56].

### 4.3. Therapeutic Strategies in Children with Epilepsy

Based on these findings, the urgent suppression of clinical seizures and EEG abnormalities, such as frontal IEDs and SBS, may be required to prevent neuropsychological impairments. During ASM selection and adjustment, physicians should strategize the therapeutic approach to controlling seizures and suppressing EEG abnormalities in children with epilepsy. Among the various ASMs, novel ASMs, such as levetiracetam and perampanel, may suppress both clinical seizures and IEDs on EEG [48,57,58,59]. These novel ASMs may represent an important addition to the treatments available for epileptic children presenting with frontal IEDs and SBS. However, it remains unclear whether seizure presence or remission are in any way related to environmental factors.

## 5. Future Perspectives

As observed in these studies, epileptic activities, including seizure severity, frequent seizures, and SE and EEG abnormalities such as frontal IEDs and SBS, can lead to reduced QOL in children with epilepsy. Better control of seizures and EEG abnormality remission may improve QOL in children with epilepsy. However, caution must be taken when generalizing the results. The early seizure and EEG abnormality remission can lead to improvements in behavioral impairments. However, improvements or reductions in other aspects, such as headache, fatigue, and stigma, using ASM treatment have not yet been identified. Studies with a larger sample size are needed to evaluate the correlation between seizure/EEG abnormality remission and improvements in these aspects in children with epilepsy. In addition, the relationship between seizures/IEDs on EEG and stigma has only been analyzed individually, without considering other biopsychosocial factors at the same time. Moreover, the mechanisms of the association between frontal IEDs and neuropsychological functioning are unclear. Therefore, the understanding of these relationships may be limited. Furthermore, there are various factors (i.e., environmental, socioeconomical, and educational factors) other than stigma that affect QOL, along with other clinical factors. However, the associations between these factors and epileptic activities have not been fully investigated. It remains unclear whether seizure presence or remission is in any way related to environmental factors. Further research is needed to clarify these aspects.

## 6. Conclusions

QOL-related factors, such as cognitive decline and behavioral disturbances, perceptions of stigma, and fatigue, are associated with epileptic activities. These disturbances are not always evident in children with epilepsy. However, severe seizures, such as frequent seizures and SE, and some forms of IEDs can lead to neuropsychological impairments. To prevent these impairments, physicians should focus on controlling seizures and suppressing EEG abnormalities as early as possible. Based on the findings from various studies, therapeutic management may be desirable to achieve seizure remission and EEG abnormalities, such as frontal IED and SBS, as early as possible to improve QOL in children with epilepsy. Among the various ASMs, novel ASMs, such as levetiracetam and perampanel, may suppress both clinical seizures and IEDs on EEG; thus, these novel ASMs may represent an important addition to the treatments available for epileptic children presenting with frontal IEDs and SBS.

## Figures and Tables

**Figure 1 jcm-13-00405-f001:**
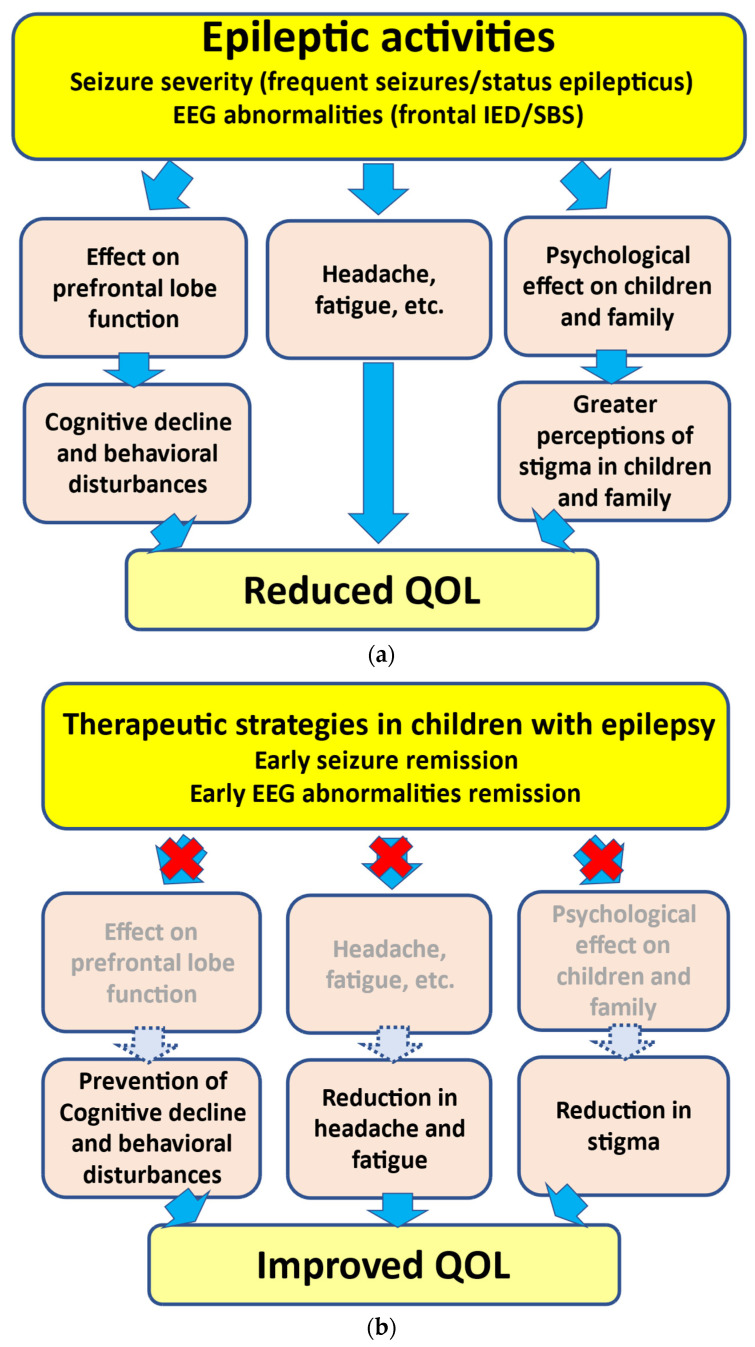
Associations between epileptic activities and reduction in QOL. (**a**) Epileptic activities including severe seizures (such as frequent seizures and SE) and EEG abnormalities (such as frontal IED and SBS) can lead to neuropsychological impairments. (**b**) Early seizure and EEG abnormality remission can lead to the prevention of cognitive decline/behavioral disturbances and a reduction in headache, fatigue, and stigma, thus resulting in an improvement in QOL. EEG, electroencephalogram; IED, interictal epileptiform discharge; SBS, secondary bilateral synchrony; QOL, quality of life.

**Table 1 jcm-13-00405-t001:** Associations between seizure severity and prefrontal lobe growth disturbances.

Epileptic Syndrome	Findings
Frontal lobe epilepsy (FLE)	# Frontal/prefrontal lobe volumes and the prefrontal-to-frontal lobe volume ratio increased serially in the drug-responsive FLE patients without cognitive decline/behavioral disturbances and non-epilepsy children. # Frontal and prefrontal lobe growth disturbances were present during the active seizure period in the refractory FLE patients with cognitive decline and behavioral disturbances. # Active seizure period was short in children with prompt growth recovery. # The growth disturbance was more severe, and the growth recovery was required a long time in children with a longer active seizure period.
Self-limited epilepsy with centrotemporal spikes (SeLECTS)	# Frontal and prefrontal lobe growth disturbances were present during the active seizure period in patients presenting atypical evolution. # The growth disturbance was more severe, and the growth recovery was required a long time in a patient with a longer active seizure period.
Self-limited epilepsy with autonomic seizures (SeLEAS)	# Frontal and prefrontal lobe growth disturbances were present after episodes of SE in SeLEAS patients presenting with behavioral disturbances. # In a patient with only one episode of SE, growth disturbance soon recovered. # Conversely, recovery of growth ratios was delayed in patients with several episodes of SE.

SE, status epilepticus.

**Table 2 jcm-13-00405-t002:** Associations between seizure severity and QOL-related factors.

QOL-Related Factors	Findings
Headache	# The frequency of seizures was 4.1 times per year in children with seizure-associated headache. # The frequency of seizures was 1.3 times per year in those with non-seizure-associated headache. # Frequent seizures may be in association with seizure-associated headache.
Fatigue	# The mean Fatigue Severity Scale scores of the children with epilepsy were significantly higher than those of the non-epilepsy children. # Frequency of seizures was sole significant clinical manifestation in association with fatigue. # A higher frequency of seizures was associated with more severe fatigue.
Stigma in children	# Children with frequent seizures presented psychosocial impairments in comparison with seizure-remission children. # Greater perceptions of stigma were in association with greater frequency of seizures.
Stigma in parents	# Parents of children with epilepsy showed significantly higher scores on the questionnaire than those of non-epilepsy children. # Greater perceptions of stigma were in association with frequency of seizures.

QOL, quality of life.

**Table 3 jcm-13-00405-t003:** Associations between QOL-related factors and IEDs on EEG.

IED on EEG as QOL-Related Factors	Findings
Seizure recurrence and IED frequency	# Seizure recurrence and extended periods of high-frequency IEDs were significantly correlated in SeLECTS.
Seizure recurrence and frontal IED	# Children with frontal IED had a significantly higher risk of seizure recurrence than those with IEDs in other regions of EEG foci.
Cognitive decline/behavioral disturbances and IED frequency	# Cognitive decline and behavioral disturbances were significantly correlated with a prolonged period of high-frequency IEDs in SeLECTS.
Cognitive decline/behavioral disturbances and frontal IED	# Cognitive decline and behavioral disturbances were significantly correlated with a prolonged period of frontal EEG focus. # A significant correlation was evident between IED frequency and ADHD-RS in children with frontal IED. # A significant correlation was seen between decreased IED frequency and reduced ADHD-RS scores after ASM treatments in children with frontal IED. # The location of IED was most commonly in the frontal region in ASD. # A correlation between decreased IED frequency and the Japanese manuals for the ABC-J score after treatment was evident in ASD children with frontal IEDs.
Cognitive decline/behavioral disturbances and SBS	# Frontal and prefrontal lobe volumes revealed growth disturbance in all EECSWS patients. # Prefrontal to frontal lobe volume ratios decreased in all EECSWS patients. # In the patients with shorter CSWS periods, the ratios were soon restored to normal values. # Prefrontal lobe growth disturbances were persistent in the patients with longer CSWS periods. # The reduction of behavioral scores was significant in EEG responders than non-responders.
Stigma in children and frontal IED	# The scores of all questions were higher in children with frontal IED. # Children presenting with frontal IED perceived a greater stigma than those with non-frontal IED.

QOL, quality of life; IED, interictal epileptiform discharge; EEG, electroencephalogram; SeLECTS, self-limited epilepsy with centrotemporal spike; ADHD—RS, attention deficit/hyperactivity disorder—rating scale; ASM, antiseizure medication; ASD, autism spectrum disorder; ABC-J, Japanese manuals for the Aberrant Behavior Checklist; SBS, secondary bilateral synchrony; EECSWS, epileptic encephalopathy with continuous spikes and waves during slow sleep.

## Data Availability

Data sharing is not applicable to this article.

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
