# Peer review of "Therapeutic Strategies in Children with Epilepsy: A Quality-of-Life-Related Perspective"

_jcm, 2024, doi:10.3390/jcm13020405_

Round 1

Reviewer 1 Report

Comments and Suggestions for Authors

This manuscript reviews the factors related to quality of life (QOL) of children with epilepsy from the multiple aspects of epilepsy including clinical and social variables. It focuses on the factors that are not only the ictal seizures but also interictal epileptiform discharges (IEDs) and cognitive and behavioral disturbances that may have a chronic impact on their QOL. The paper along with the conceptual figure is well elaborated with conceptual illustrations and makes an important contribution to the QOL literature in children with epilepsy.

A few suggestions and questions follow below.

1) Introduction

The beginning of the section (Line37-57) seems to have an abundance of an overview of QOL regarding epileptic seizures, IEDs, cognitive impairment, and stigma. There are citations to support the subsequent section on fatigue and beyond, but not in the section above.  Adding key citations for each factor might make for a stronger description.

It may also be more structured and clear to describe the purpose of this whole review manuscript at the end of the Introduction section, in order to go into each section below. 

When structuring the manuscript, it may be helpful to have a theoretical model of the relationship among biopsychosocial aspects of children with epilepsy. For example, biopsychosocial model (Engel, 1977) may be one example to conceptually organize the QOL-related factors. This may be related to Figure 1. 

2)2. Can seizure severity lead to reduced QOL in childhood epilepsy?

2.1. Seizure severity and..

The effect of seizures seems to focus mainly on frontal lobe and its cognitive effect. I understand the author is an expertise in this area with many citations of the author. It maybe helpful to review other parts of brain regions or various types of epilepsy comprehensively that may affect cognitive and behavioral functionings.  

2.4, 2.5  stigma

These two sections discuss self-stigma of children and stigma of parents. While these are counted as psychological and social factors. You may also consider a critical analysis of the cited literatures. For example, if the relationship between seizure and stigma is only analyzed individually without considering other biopsychosocial factors simultaneously (i.e., multiple regression analysis) , these relationship may have limitation in its statement.

In addition, it is unclear how this manuscript places stigma in between QOL and clinical factor. Are you considering stigma as one indicator of QOL; or one psychological factor that affects QOL along with seizure factor? Generally in the literature of QOL or psychological theories, QOL is considered as the final outcome of biopsychosocial factors.

3. Can EEG abnormalities lead to...

3.3. It is very interesting to know the IED might play a role in neuropsychological functions or stigma, which is more of a psychological function. However, it may be helpful to deepen the review if the mechanism of these associations are described (if not, describe as a limitation of the current literature).

5. Future Perespectives

You may elaborate more on the limitation of the current literature as there are more than stigma (i.e., environmental factor, socioeconomical factor, educational factor) that affect QOL along with other clinical factors.

Author Response

I thank the reviewer for the insightful and constructive comments. My responses to Reviewer #1 are as those with attached file.

Reviewer 2 Report

Comments and Suggestions for Authors

I have carefully reviewed the manuscript titled "Therapeutic strategies in children with epilepsy - from the point of view of QOL-related factors" submitted by Hideaki Kanemura. Unfortunately, I find significant issues with the overall content, presentation, and language quality. Here are my detailed comments:

  • The abstract lacks a clear structure and concise presentation of key points. It is challenging to follow the flow of the content, and there is an abundance of information that could be organized more effectively.
  • The manuscript contains redundant statements and repetitive information. It is important to streamline the content and present information more concisely.
  • The manuscript covers a wide range of topics related to epilepsy, seizure severity, and EEG abnormalities, making it challenging for the reader to identify a central theme or focus of the study.
  • The language used is convoluted and unclear. There are numerous grammatical errors, and some sentences are difficult to understand. The abstract needs significant improvement for clarity and readability.
  • Specific details about the therapeutic strategies are lacking. The abstract mentions the importance of therapeutic management but does not provide concrete details about the proposed strategies.

I recommend restructuring the manuscript to provide a clear focus, eliminating redundancy, and improving language and grammar. Additionally, please consider providing more specific details about the proposed therapeutic strategies.

Comments on the Quality of English Language

It needs improvement

Author Response

I thank the reviewer for the insightful and constructive comments. My responses to Reviewer #2 are as those with attached file.

Round 2

Reviewer 1 Report

Comments and Suggestions for Authors

The author's willingness to thoroughly respond to the reviewer's points is commendable.

There are several major modifications that should be reconsidered;

(Figure 1-2) This framework can be revisited based on the biopsychosocial model. Biological aspects include 1) effects on prefrontal functions, 2) headaches, fatigue, etc., and 3) psychological effects on children and families belong to the psychological aspects, which goes without saying. Thus, the order of the three boxes is suggested as 1) → 2) → 3). Similarly, in the second box, headache and fatigue belong to the biological aspect and should be in one box, while stigma should be in another.

One factor missing from the biopsychosocial model is the environmental and social factors. The question arises as to whether seizure presence or remission is in any way related to environmental factors.

Comments on the Quality of English Language

There are some minor grammatical errors to review throughout the manuscript, mainly in the revised section.

Author Response

I thank the reviewer for the insightful and constructive comments. My responses to Reviewer #1 are as follows.

  • In Figure 1-2: This framework can be revisited based on the biopsychosocial model. Biological aspects include 1) effects on prefrontal functions, 2) headaches, fatigue, etc., and 3) psychological effects on children and families belong to the psychological aspects, which goes without saying. Thus, the order of the three boxes is suggested as 1) 2) 3). Similarly, in the second box, headache and fatigue belong to the biological aspect and should be in one box, while stigma should be in another.

I thank the reviewer for meaningful suggestions. In accordance with reviewer’s suggestion, I have modified the Figure 1 and 2.

  • One factor missing from the biopsychosocial model is the environmental and social factors. The question arises as to whether seizure presence or remission is in any way related to environmental factors.

I thank the reviewer for meaningful suggestions. As reviewer suggested, the environmental and social factors are missed from the biopsychological model. Thus, the question arises as to whether seizure presence or remission is in any way related to environmental factors. However, I regret that it remains unclear whether seizure presence or remission is in any way related to environmental factors from our previous investigations. In response the reviewer’s comment, I have added the sentences in Introduction section as follows (lines 354–355 and 391-392 of the revised manuscript):

“However, it remains unclear whether seizure presence or remission is in any way related to environmental factors.” (line 354-355)

“It remains unclear whether seizure presence or remission is in any way related to environmental factors.” (lines 391-392)

  • There are some minor grammatical errors to review throughout the manuscript, mainly in the revised section.

In response to the comments from the reviewer, the revised manuscript has been re-edited by a native English speaker again. Article title, several phrases, and sentences have thus been changed. I have added the sentences as follows (lines 411–413 of the revised manuscript):

“The author thanks FORTE Science Communications (https://www.forte-science.co.jp/) and MDPI AUTHOR services (https://www.mdpi.com/authors/english.) for the English language editing.”

In addition, the revised manuscript has been re-edited by a native English speaker again. I have also sent the English Editing Certificate with attached file.

Reviewer 2 Report

Comments and Suggestions for Authors

The manuscript still has improvement chances

Comments on the Quality of English Language

Moderately editing required

Author Response

I thank the reviewer for the insightful and constructive comments. My responses to Reviewer #2 are as follows.

  • The manuscript still has improvement chances.

I thank the reviewer for meaningful comments. I have modified the Figure 1 and 2. In addition, the revised manuscript has been re-edited by a native English speaker again. Article title, several phrases, and sentences have thus been changed. I have added the sentences as follows (lines 411–413 of the revised manuscript):

“The author thanks FORTE Science Communications (https://www.forte-science.co.jp/) and MDPI AUTHOR services (https://www.mdpi.com/authors/english.) for the English language editing.”

In addition, I have also sent the English Editing Certificate with attached file.

Round 3

Reviewer 1 Report

Comments and Suggestions for Authors

Although the relationship among these factors still needs a lot of research and perhaps more review process, I appreciate and commend the authors' thorough efforts to improve the manuscript.